# Effects of Incorporating B-Tricalcium Phosphate with Reaction Sintering into Mg-Based Composites on Degradation and Mechanical Integrity

Kai Narita [1],*,[†], Sachiko Hiromoto [2], Equo Kobayashi [1] and Tatsuo Sato [3]

[1] Department of Metallurgy and Ceramics Science, Tokyo Institute of Technology, 2-12-1 Ookayama, Meguro-ku, Tokyo 152-8552, Japan; equo@mtl.titech.ac.jp
[2] Corrosion Property Group, Research Center for Structural Materials, National Institute for Materials Science, 1-2-1 Sengen, Tsukuba 305-0047, Japan; hiromoto.sachiko@nims.go.jp
[3] Tokyo Institute of Technology, Tokyo 152–8552, Japan; sato.tatsuo8@gmail.com
* Correspondence: kai.y.narita@gmail.com
† Current address: Division of Engineering and Applied Sciences, California Institute of Technology, Pasadena, CA 91125, USA.

**Abstract:** For applications of biodegradable load-bearing implants, we incorporated 10 or 20 vol% β-tricalcium phosphate (β-TCP) into Mg-based composites through reaction sintering in the spark plasma sintering process. We previously reported that the evolved microstructure enhanced mechanical properties before degradation and modified in vitro degradation behaviors. In this study, immersion tests in physiological saline and subsequent compression tests in the air were conducted to investigate the effects of degradation on mechanical integrity. In the immersion tests, Mg/β-TCP composites showed no visible disintegration of sintered particles due to interfacial strength enhanced by reaction sintering. Local corrosion was observed in the Mg matrix adjacent to the reaction products. In addition, Mg/10% β-TCP showed dense degradation products of Mg(OH)$_2$ compared with Mg and Mg/20% β-TCP. Those degradation behaviors resulted in reducing the effective load transfer from the Mg matrix to the reaction products as reinforcement. The yield strength decreased by 18.1% for Mg/10% β-TCP and 70.9% for Mg/20% β-TCP after six days of immersion. These results can give a broad view of designing spark plasma sintered Mg/bioceramic composites with the consideration of mechanical integrity.

**Keywords:** Mg-based composites; β-tricalcium phosphate; degradation behaviors; mechanical integrity; microgalvanic corrosion

## 1. Introduction

Biodegradable load-bearing implants are of great interest in orthopedic fields as such devices do not require surgery for their removal after the fractured bones have healed, therefore reducing the physical burden on patients [1]. Magnesium (Mg)-based materials are considered to be good candidates for biodegradable load-bearing implants and their applicability has been broadly explored. Mg provides an exceptional combination of strength and ductility: Mg is more ductile than ceramic biomaterials such as hydroxyapatite [2] and is stronger than biodegradable polymers including poly-L-lactic acid (PLLA) and poly(lactic-co-glycolic) acid (PLGA) [3]. The elastic modulus of Mg is close to human cortical bone, thereby reducing the stress-shielding effect [2,4]. In addition, Mg is essential to human metabolism and may have stimulatory effects for the growth of new bone [5–8]. Despite these advantages, a significant complication arises as Mg can degrade too quickly in the high-chloride environment of the physiological system, which can deteriorate the mechanical integrity before the bone has sufficiently healed [9]. Furthermore, the initial strength of unalloyed Mg before degradation exhibits 65–100 MPa of compressive yield strength, which is insufficient to be applied to load-bearing conditions compared with the

strength of currently used metallic biomaterials such as titanium alloys (758–1117 MPa) and human cortical bone (130–180 MPa) [2].

In order to enhance initial strength and corrosion resistance, novel Mg alloys and Mg-based composites have been explored [10,11]. Utilizing advantages of the biocompatibility of bioceramics, magnesium/bioceramic composites have been developed using hydroxyapatite (HA), β-type tricalcium phosphate (β-TCP), calcium polyphosphate (CPP), and other types of bioceramics [12–15]. Among those bioceramics, β-TCP is a promising candidate for the Mg-based composites due to its biocompatibility, bioactivity, and osteoconductivity, which have already led to its current applications in medical fields [16]. In addition, β-TCP leads to completely biodegradable Mg-based composites in contrast to HA, which is not biodegradable. Therefore, pure Mg or Mg alloy/β-TCP composites have been fabricated in various methods including suction casting, hot extrusion, and the high shear solidification (HSS) technique followed by equal-channel angular extrusion (ECAE) [17–21]. Among those fabrication methods on Mg/bioceramic composites, mechanical properties were improved by composite strengthening when strong interfacial bonding between matrix and reinforcement was evolved during the fabrication process, whereas Mg/bioceramic composites produced by suction casting or powder metallurgy were likely to have lower mechanical properties than pure Mg or Mg alloy [12,18,22,23].

We have proposed the use of advantageous reactions to produce Mg-based composites in order to achieve sufficient interfacial bonding for composite strengthening [24,25]. Spark plasma sintering (SPS) was chosen for the sintering method because it enables high densification due to the vacuum condition, external pressure, and the use of direct current to generate heat. The SPS method has been employed for the preparation of particulate-reinforced Mg-based composites [26–29] and biodegradable Mg/bioceramic composites [30–33]. In our previous study, the Mg/β-TCP composites with nearly 100% relative density and strong interfacial bonding between Mg matrix and the reaction products were developed by taking advantage of the reactions between Mg and β-TCP during sintering [24]. The developed Mg/β-TCP exhibited higher mechanical properties than Mg in the same fabrication scheme due to composite strengthening with sufficiently strong interfacial bonding for load transfer [24]. Without the reactions, the Mg/β-TCP composites showed lower strength than sintered Mg because β-TCP did not act as reinforcement due to insufficient interfacial bonding between β-TCP and Mg for load transfer, as other Mg/bioceramic composites showed deteriorated strength [12,22,23]. We also conducted in vitro degradation tests using revised simulated body fluid and cytocompatibility tests on the developed Mg/β-TCP composites [34]. Among the tested samples, sintered Mg, Mg/10% β-TCP, and Mg/20% β-TCP, Mg/10% β-TCP showed the slowest $Mg^{2+}$ ion release rate after the immersion for 11 days as a result of the counter-effects between microgalvanic corrosion and deposition of Ca-P containing degradation layers, both of which occurred due to the existence of the unique microstructure evolved by the reactions between Mg and β-TCP [34].

With considerations for future application, in the present work we have investigated effects of the degradation on mechanical integrity of the Mg/β-TCP composites sintered by the SPS. Fewer studies addressed the mechanical integrity of Mg-based materials compared with ones focusing on the corrosion resistance and initial mechanical properties [10,35–38]. Jie Zhou's group showed that incorporating 20 vol% bredigite particles into the Mg matrix by pressure-assisted sintering improved mechanical integrity upon the immersion in a cell culture medium compared with monolithic Mg [39]. They showed that corrosion pits acted as nucleation sites of mechanical cracks for Mg/bredigite composites. In contrast to his fabrication method without reactions, our preparation method utilizing reactions produced a distinct microstructure having reaction products, from which would arise different degradation behaviors and resultant mechanical integrity. Therefore, this work may not only reveal the capabilities of Mg/β-TCP composites but also give a broad view of designing Mg/bioceramic composites with the consideration of mechanical integrity.

In this paper, degradation behaviors and the resultant strength change of the Mg/β-TCP composites were examined by compression tests following the immersion tests in physiological saline. Physiological saline is the simplest simulated physical solution which does not contain calcium ions or phosphate ions, unlike other common simulated physical solutions such as Kokubo solution [40]. As the first evaluation of mechanical integrity of Mg/β-TCP composites fabricated by the SPS, a saline solution was chosen to investigate degradation behaviors and mechanical integrity without complicated effects caused by deposition due to the presence of calcium ions and phosphate ions in the solution [41]. The effect of the β-TCP fraction on the degradation properties in the microstructure evolved by reactions during the SPS was also discussed.

## 2. Materials and Methods

Pure Mg powder classified by a 180 μm sieve, and 10 or 20 vol% β-TCP powder with an approximate size of 1–2 μm were used in this study. The Mg particles show an angular shape (see Figure S1 in the Supplementary Materials). We measured Mg particle size by optical microscope and showed the size distribution of Mg powder in Figure S2 in the Supplementary Materials. The measured particle size ranged from 80 μm to 320 μm. The average particle size was 168 μm. The purity of the Mg powder was 99.5%, and impurity composition was tabulated in Table S1 in the Supplementary Materials. The impurity concentrations were lower than the levels above which the corrosion rate increases dramatically [42]. We mixed Mg powder and β-TCP powder agents under an Ar atmosphere by planetary ball milling with zirconia balls with a diameter of 1 mm in a stainless steel container at 100 rpm for 12 h. The weight ratio of ball/powder was 1:10. As a control, Mg powder without β-TCP underwent the same processes.

Preliminarily pressure of 25 MPa was applied on the mixed powder in a graphite die with an inner diameter of 15 mm, and then the green compact was sintered for 10 min under vacuum using the SPS at 793 K and external pressure of 50 MPa. The SPS parameters were chosen to cause the sintering reactions and densification of Mg/β-TCP composites [24]. Sintered Mg, as a control, was produced in the same manner as Mg/β-TCP composites using pure Mg powder without adding β-TCP. The sintered samples were machined into 4 mm × 4 mm × 7.5 ($\pm$0.5) mm rectangular cuboid shape specimens. The samples were observed by optical microscopy (Leica Microsystems, Wetzlar, Germany) after polishing with abrasive papers up to 4000 grit and 3 μm diamond particles. To discern boundaries where Mg particles were sintered (hereinafter, called "particle boundaries"), the sintered Mg was etched by a Nital solution. The boundaries of Mg particles of the sintered Mg were characterized by Auger Electron Spectroscopy (AES) (JEOL, Tokyo, Japan).

The machined specimens were ground with abrasive papers up to 4000 grit, and rinsed with ethanol; then each specimen was immersed in 50 mL of physiological saline (0.9 mass% NaCl solution) at 310 K. We chose physiological saline to do an accelerated and simplified corrosion test, which is likely to show degradation and strength changes by corrosion attacks by chloride ions in a relatively short time and eliminate complex effects of calcium and phosphate ions in media on degradation. Note that physiological saline contains higher chloride ion concentrations than human blood plasma and other corrosion media such as simulated body fluids (SBF) [40]. The ratio of the total surface area of the sample to solution volume was about 40 mL/cm$^2$. Three specimens for each prescribed immersion period (1, 3, 6, or 9 days) were prepared for each type of sample: Mg/10% β-TCP, Mg/20% β-TCP, and sintered Mg. In the initial 2.5 h of immersion, the pH change was measured in situ using a digital pH meter. At the end of the prescribed immersion period, each specimen was removed from the physiological saline, and the pH of the physiological saline was measured using a digital pH meter. The specimens were rinsed by ethanol and then dried under a vacuum for more than 30 min. The dried specimens were photographed and weighed. Mass change of the specimens was calculated according to the following equation, $M/M_0 \times 100$ (%), where $M_0$ and $M$ are mass of the specimens before and after immersion and drying, respectively. The dried specimens after

immersion were observed by optical microscopy (Leica Microsystems, Wetzlar, Germany) and characterized by X-ray diffraction (XRD) (Rigaku, Tokyo, Japan). XRD was conducted using CuKα radiation with 30 mA current and 40 kV voltage. In addition, degradation products on the dried specimens after three days of immersion were coated by a 5 nm thick layer of gold and then characterized by scanning electron microscopy (SEM) (JEOL, Tokyo, Japan) and attached energy dispersive spectroscopy (EDS) (JEOL, Tokyo, Japan).

A compression test in ambient air was conducted using the specimens before and after the immersion test. The specimens after immersion were prepared as described in the previous section. The compression tests were conducted with the specimens which can be handled without fracture after immersion (i.e., the specimens after immersion up to one day for the sintered Mg, nine days for the Mg/10% β-TCP, and six days for the Mg/20% β-TCP). The crosshead speed was 0.2 mm/min and the displacement was measured by a video extensometer (Keyence, Osaka, Japan). The stress was calculated using the cross-sectional area of the specimens measured before immersion. The compression test was conducted in triplicate for each type of sample and immersion period. After the compression test, cracks on the exterior surface were observed by SEM (JEOL, Tokyo, Japan).

## 3. Results

Figure 1 illustrates the microstructure of the sintered Mg and Mg/ β-TCP composites before the immersion test. The sintered Mg showed boundaries of the sintered particles due to the etching using a Nital solution. The Mg particles were aligned vertically to the pressing direction (lateral direction in Figure 1) in the SPS process. Mg/ β-TCP composites showed the Mg matrix as bright regions and reaction products (hereinafter, called "RPs") of which the main component was MgO [24] as dark regions. The total area of RPs in the Mg/20% β-TCP was larger than that in the Mg/10% β-TCP. Figure 1D,E show scanning electron microscope (SEM) images and the corresponding energy dispersive spectroscopy (EDS) analysis (reproduced from our previous study [34]). The results show that Ca was diffused to the adjacent area of the Mg matrix to the RPs area. The detection of carbon was attributed to adventitious carbon contamination and diffusion from a carbon die used in the SPS process [34].

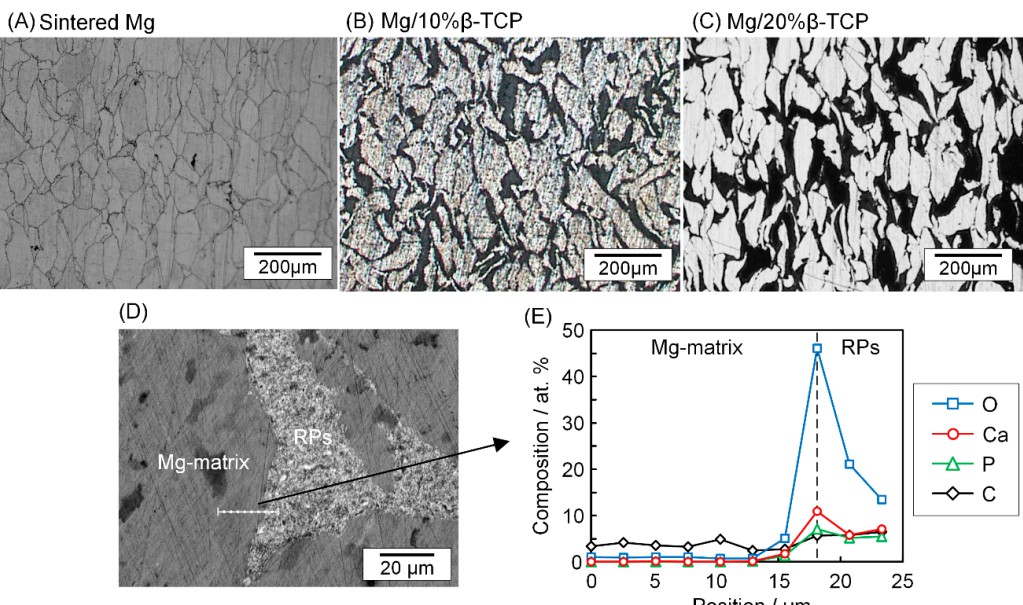

**Figure 1.** Optical micrographs of the (**A**) sintered Mg, (**B**) Mg/10% β-TCP, (**C**) Mg/20% β-TCP before immersion, (**D**) SEM image of the Mg/10% β-TCP showing the interface between Mg matrix and reaction products (RPs) area, and (**E**) line analysis of EDS across the interface. The balanced concentration of Mg was not shown in (**E**) (**D** and **E** are reproduced from our previous work [34], copyright permission obtained).

The formed RPs were characterized by XRD as shown in Figure 2. The Mg/10% β-TCP and Mg/20% β-TCP showed Mg, β-TCP, and MgO peaks. The peak intensities increased as increasing β-TCP fractions. The peak positions of MgO for Mg/10% β-TCP and Mg/20% β-TCP were shifted to low angles compared with that for sintered Mg. The sintered Mg also showed the MgO peak. Native oxide layers of Mg particles and oxygen residue in the SPS process may form MgO along the boundaries of Mg particles. This is supported by our observations of high oxygen concentrations on the boundary of sintered Mg particles (Figure S3 in the Supplementary Materials). The small peaks at 35.4 and 46.2 were possibly derived from $Ca(PO_3)_2$.

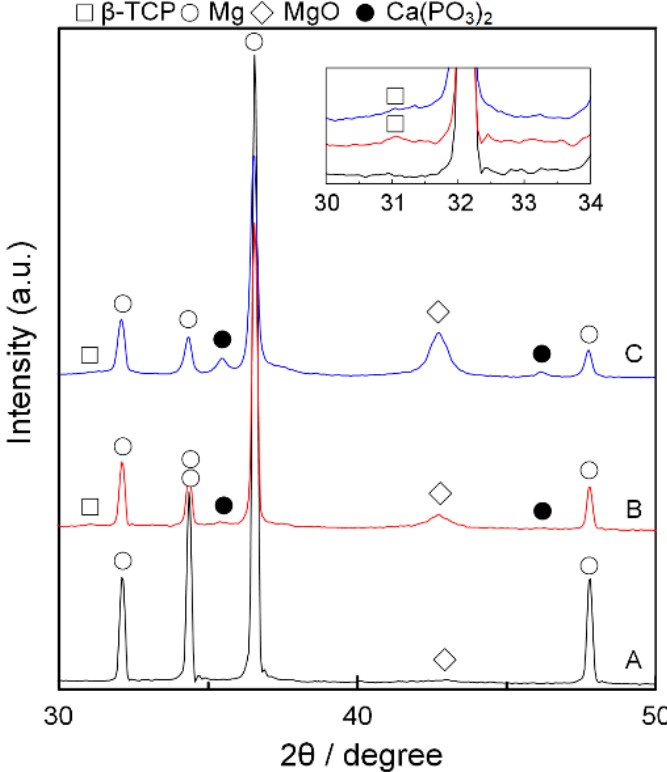

**Figure 2.** XRD patterns of (**A**) the sintered Mg, (**B**) Mg/10% β-TCP, (**C**) Mg/20% β-TCP before immersion. Inset shows magnified patterns with small peaks of β-TCP.

Figure 3 shows macrographic images of the specimens before and after the immersion test. The white degradation products covered the sintered Mg after one day of immersion and became thicker after three days. Large cracks were observed perpendicular to the pressing direction in the SPS process (i.e., longitudinal direction). Some specimens of sintered Mg after three days of immersion broke apart due to the large cracks. In contrast, the Mg/10% β-TCP and Mg/20% β-TCP showed the locally deposited white degradation products and the area covered by the white degradation products increased as the immersion period increased. After six days of immersion, some particles detached from the bottom corner of the Mg/20% β-TCP specimen. Since the sintered Mg was fractured after three days of immersion and the Mg/20% β-TCP showed fragmentation after six days of immersion, immersion tests were not conducted for longer periods with the sintered Mg and Mg/20% β-TCP. After nine days of immersion for the Mg/10% β-TCP, the white degradation products covered its entire surface, and no fragmentation occurred.

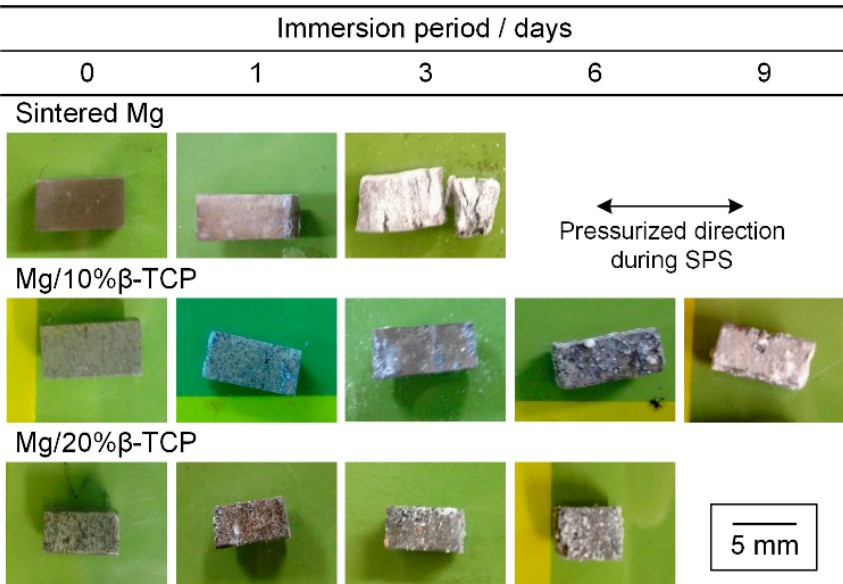

**Figure 3.** Photographs of specimens before and after immersion for each prescribed period.

Figure 4 shows the degraded surface after the immersion test. Pits with less than 10 μm diameter, as shown in Figure 4(A2) were observed on the entire surface of the sintered Mg after the one day of immersion. On the corner of the specimen shown in Figure 4(A1), cracks induced by degradation were observed along the Mg particle boundaries. After three days of immersion, the surface of the sintered Mg was fully covered by white degradation products and showed large cracks (Figure 4B). The large cracks in Figure 4B corresponded to the visible cracks of the sintered Mg in Figure 3. The white degradation products were also deposited inside the large cracks.

The Mg/β-TCP composites showed different degradation behaviors from the sintered Mg. No cracks were observed after immersion. In the Mg matrix, degradation occurred inhomogeneously and metallic luster in some parts was observed until six days for the Mg/10% β-TCP and three days for the Mg/20% β-TCP. Figure 4(C2,D2,G2,H2) show local corrosion including pitting in the Mg matrix adjacent to the RPs area. Interestingly, Mg/20% β-TCP showed more prominent local corrosion in the Mg matrix region adjacent to the RPs area and fewer pits in the inner Mg matrix region compared with the Mg/10% β-TCP (Figure 4(C2,G2)). Translucent white degradation products were observed mainly on the RPs area and the corroded Mg matrix area. Eventually, almost the entire surface was subject to degradation and covered by the degradation products after nine days for the Mg/10% β-TCP and six days for the Mg/20% β-TCP.

Figure 5 shows in situ pH change of the physiological saline for the initial 2.5 h of immersion. The pH of saline with the Mg/β-TCP composites increased rapidly in the beginning and reached slowly around 10, in contrast to a gradual increase for the sintered Mg. In particular, the Mg/20% β-TCP exhibited more rapid pH increment than the Mg/10% β-TCP. Furthermore, pH measurement after the immersion for any prescribed periods (1, 3, 6, and 9 days) showed stabilized pH, around 10.5, for all the samples.

XRD patterns after the immersion test are shown in Figure 6. All the samples showed $Mg(OH)_2$ peaks in addition to the peaks which were seen before immersion. As the immersion time increased, the intensity of Mg, MgO, and $Ca(PO_3)_2$ peaks decreased while the intensity of $Mg(OH)_2$ became increased.

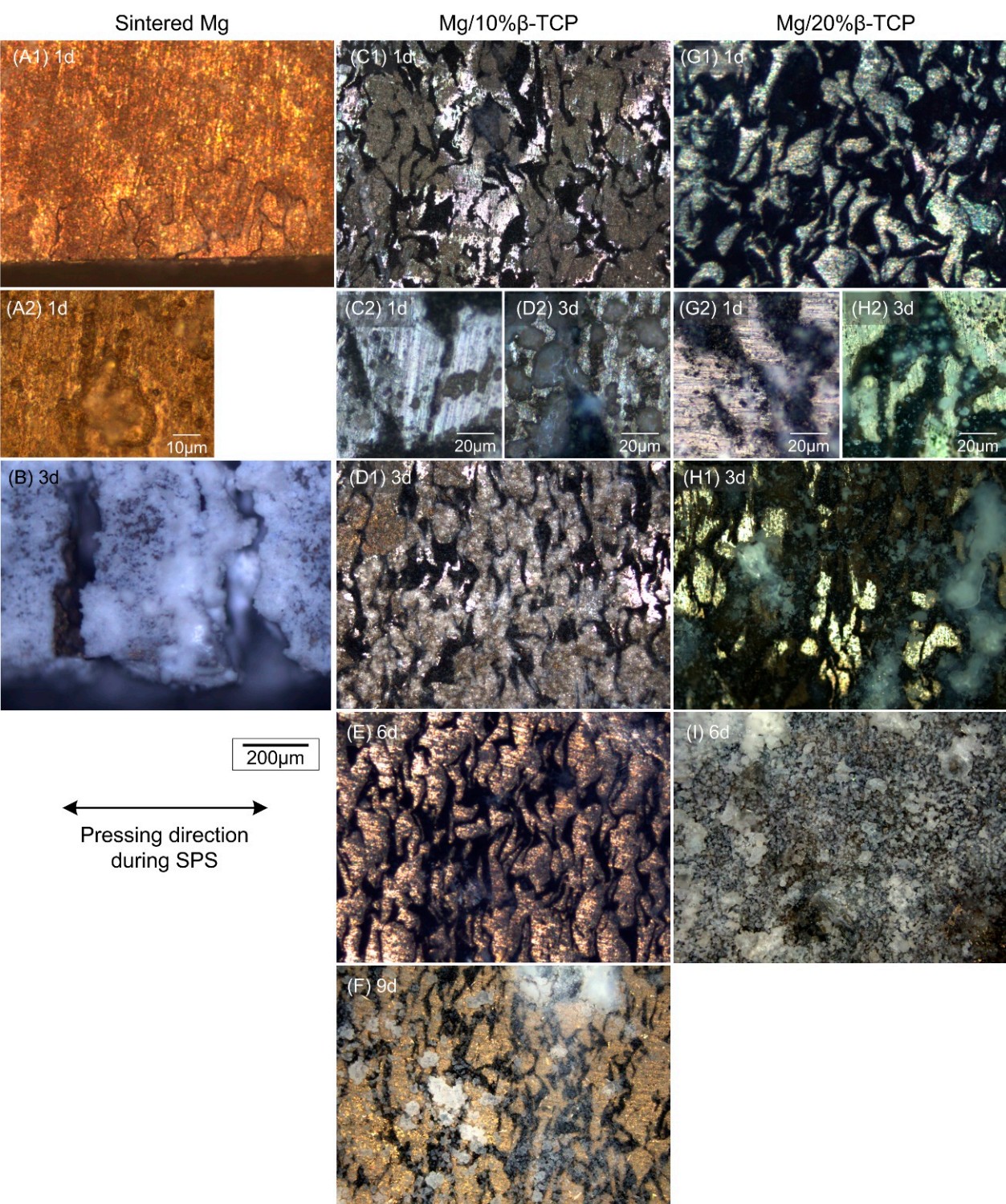

**Figure 4.** Optical micrographs of the (**A1**) sintered Mg after immersion for 1 day and (**B**) 3 days; (**C1**) Mg/10% β-TCP after immersion for 1 day, (**D1**) 3 days, (**E**) 6 days, and (**F**) 9 days; (**G**) Mg/20% β-TCP after immersion for 1 day, (**H1**) 3 days, and (**I**) 6 days. (**A2**,**C2**,**D2**,**G2**,**H2**) high-magnification images of (**A1**,**C1**,**D1**,**G1**,**H1**), respectively.

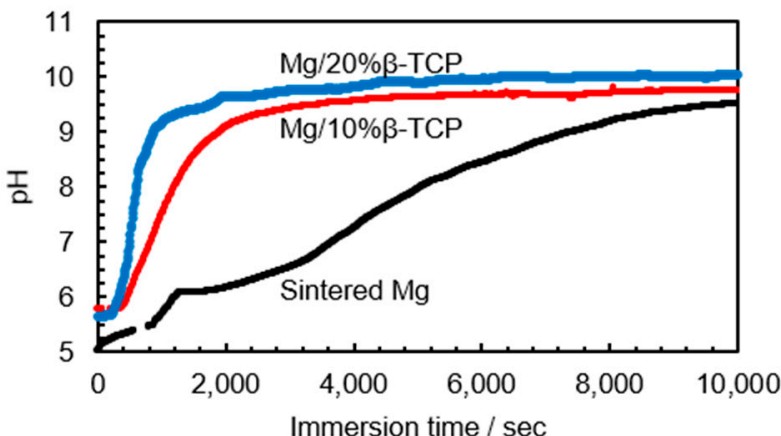

**Figure 5.** Changes of pH values in the initial 2.5 h after immersion in physiological saline.

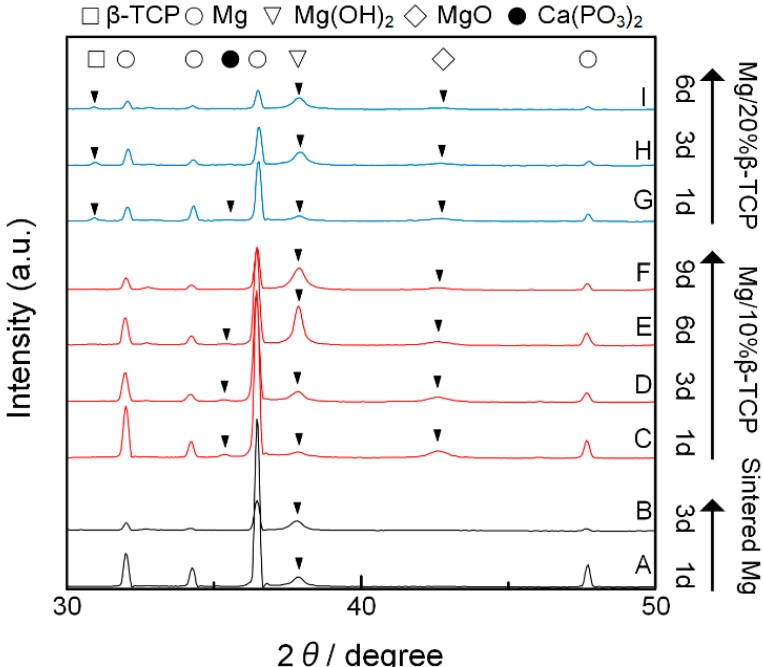

**Figure 6.** XRD patterns of samples of the (**A**) sintered Mg after immersion for 1 day and (**B**) 3 days; (**C**) Mg/10% β-TCP after immersion for 1 day, (**D**) 3 days, (**E**) 6 days, and (**F**) 9 days; (**G**) Mg/20% β-TCP after immersion for 1 day, (**H**) 3 days, and (**I**) 6 days. Peaks on each pattern were labeled by solid triangles except for Mg, which is seen in all patterns.

SEM images in Figure 7 show the degradation product morphology after three days of immersion. The sintered Mg showed flower-like degradation products composed of thin plates with thickness ranging from 0.1 μm to 0.5 μm (Figure 7A). The Mg/10% β-TCP showed degradation product morphology similar to the flower-like structure of the sintered Mg (Figure 7B). Compared with the sintered Mg, the plates of the flower-like degradation products were located more closely to one another and gaps between the plates were significantly smaller. Interestingly, the Mg/20% β-TCP showed two different morphologies of degradation products: triangle-shaped degradation products (Figure 7C) and flower-like degradation products composed of thin round plates (Figure 7D).

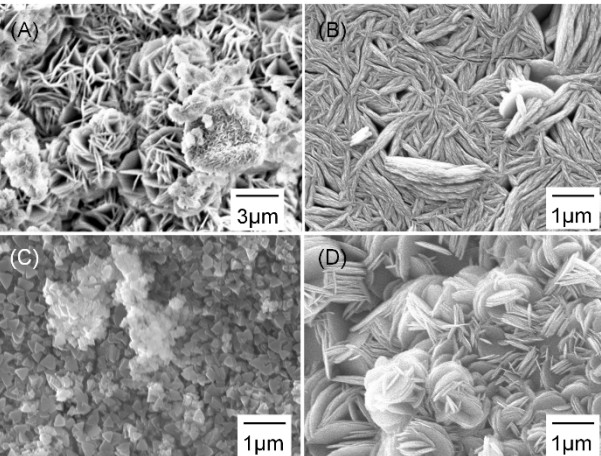

**Figure 7.** SEM images of the surface after immersion for 3 days showing degradation products on the (**A**) sintered Mg, (**B**) Mg/10% β-TCP, and (**C,D**) different regions in Mg/20% β-TCP.

Furthermore, the EDS analysis on degradation products detected mainly Mg and O with O/Mg molar ratio of 1.96 for the sintered Mg (Figure 7A), 1.87 for Mg/10% β-TCP (Figure 7B), 1.98 for the triangle-shaped products on Mg/20% β-TCP (Figure 7C), and 2.42 for the flower-like products on Mg/20% β-TCP (Figure 7D). The high O/Mg molar ratio for the region in Figure 7C may be partially due to the presence of β-TCP underneath. Ca and P were detected for several atomic percentages for both types of degradation products on Mg/20% β-TCP.

Figure 8 shows mass change after the immersion test. The sintered Mg showed a remarkable mass increase by 21.6% after three days of immersion. In contrast, the mass of the Mg/10% β-TCP was stable until six days and then decreased slightly. The Mg/20% β-TCP showed a marginal mass increase after one day of immersion and then a subsequent decrease after three days of immersion. The mass nearly stabilized between three and six days.

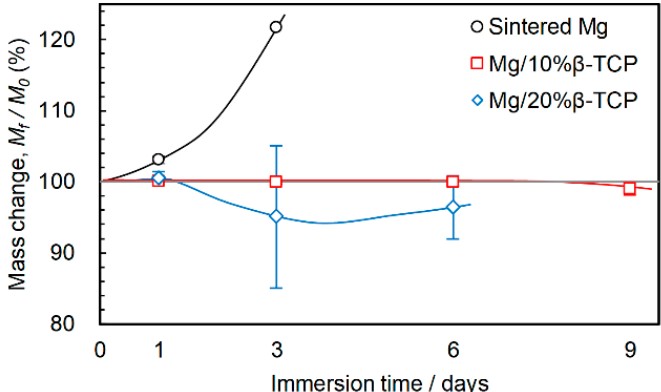

**Figure 8.** Mass change after immersion for prescribed periods.

Figure 9A,B show changes in the maximum compressive stress and 0.2% yield stress as the immersion period increased. Before immersion, the Mg/ β-TCP composites showed considerably enhanced initial strength compared with the sintered Mg. The Mg/20% β-TCP exhibited a maximum compressive strength two times higher and 0.2% yield strength four times higher than those of the sintered Mg.

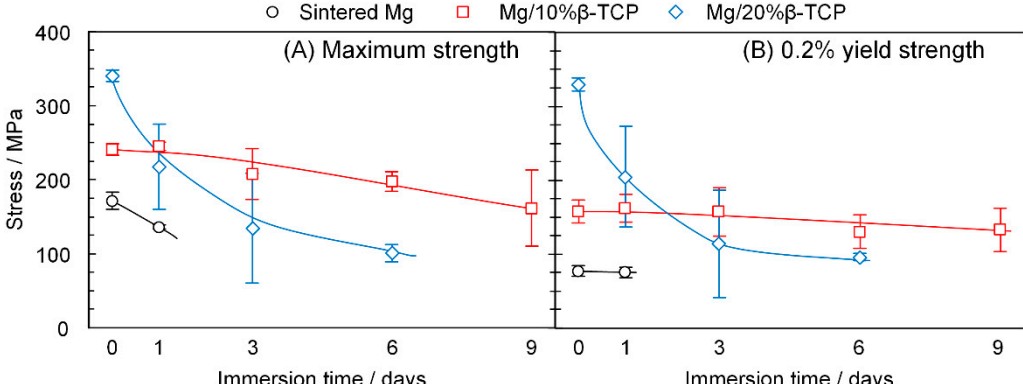

**Figure 9.** (**A**) Maximum strength and (**B**) 0.2% yield strength before and after immersion for each prescribed period.

After one day of immersion, the mean maximum strength of the sintered Mg decreased from 171.3 MPa to 135.9 MPa, whereas the yield strength was stable up to one day. The Mg/10% β-TCP decreased the maximum strength gradually to 161.4 MPa after nine days of immersion. The yield strength was stable up to three days, and then reduced marginally after six and nine days of immersion. On the other hand, the Mg/20% β-TCP exhibited a significant reduction of not only the maximum strength but also yield strength after three days of immersion. Then, both yield and maximum strengths slightly decreased by the immersion from three days to six days.

Figure 10 shows the observation of cracks after the compression tests with as-sintered and as-immersed samples. In the sintered Mg, both a large crack propagating across the specimen and small cracks were observed along the particle boundaries for as-sintered and as-immersed samples. On the other hand, as-sintered and as-immersed samples of the Mg/10% β-TCP showed cracks mainly in the RPs area (seen as a relatively bright region) located between sintered Mg particles. The as-sintered Mg/20% β-TCP also contained cracks propagating in the RPs area dominantly as well as inside the Mg matrix area. For the as-immersed Mg/20% β-TCP, cracks were observed inside the RPs area. The fragments on the surface consisted of the cracked degradation layer.

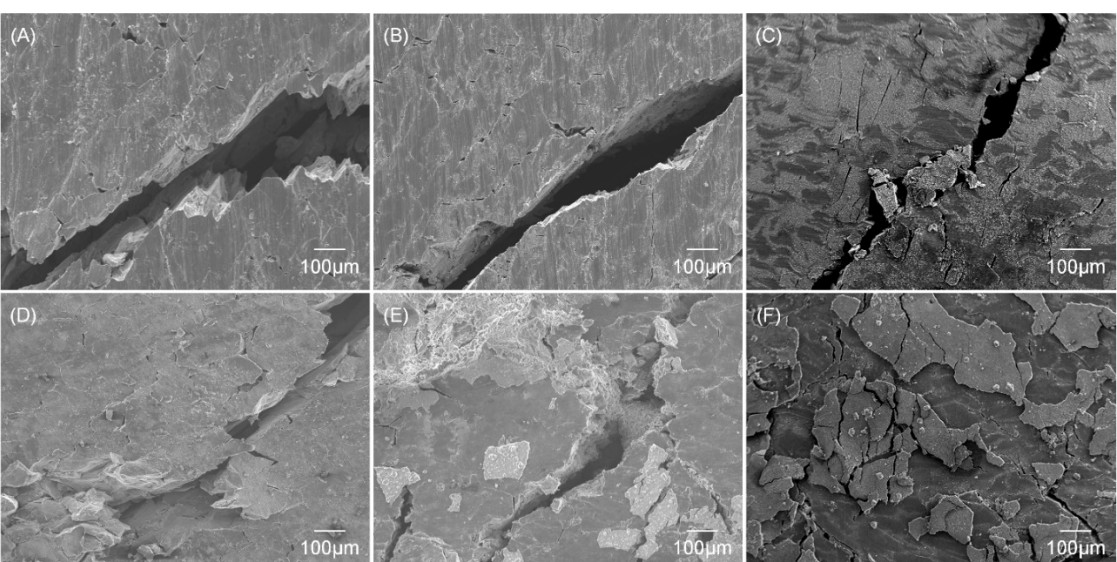

**Figure 10.** SEM images of fracture surface after compression tests for non-immersed samples of the (**A**) sintered Mg, (**B**) Mg/10% β-TCP, and (**C**) Mg/20% β-TCP, and samples of the (**D**) sintered Mg after 1 day immersion, (**E**) Mg/10% β-TCP after 9 days immersion, and (**F**) Mg/20% β-TCP after 6 days immersion.

## 4. Discussion

### 4.1. Microstructure Evolution of Mg/β-TCP Composites during Sintering with Reactions

As our previous study discussed, the microstructure of the Mg/10% β-TCP and Mg/20% β-TCP had various components due to unique processes involving reactions between Mg and β-TCP during the SPS [24]. The detailed analyses of microstructural evolution were conducted and described elsewhere [24]; a brief description is given here. During the SPS process at 793 K with the compression, the reaction between Mg and β-TCP involved Ca solid solution diffusion from β-TCP into Mg, which lowered the melting point of Mg below 793 K and formed a locally melted region surrounding the reacted β-TCP. The locally melted Mg–Ca system flowed into the gap of agglomerated β-TCP particles with the help of applied pressure in the SPS process. The infiltrated Mg–Ca liquid system continuously reacted to other β-TCP particles in the interior region of the agglomerations, which eventually produced almost fully densified composites and strong interfacial bonding between Mg and the RPs in spite of large agglomerations of additive powder [24]. The reactions produced MgO as a major component and $Ca(PO_3)_2$ as a minor, as confirmed by XRD (Figure 2). Since both MgO and $Ca(PO_3)_2$ are biodegradable and biocompatible, the Mg/β-TCP composites containing these RPs can act as biodegradable materials [43–45]. Other compounds such as CaO or Mg–Ca intermetallic compound, $Mg_2$Ca, were not detected by XRD. The substitution of Ca in MgO was confirmed by peak shift of MgO to low angles in Mg/10% β-TCP compared with the sintered Mg. In summary, Mg/β-TCP composites had a microstructure composed of Mg matrix, β-TCP, RPs, and the Mg–Ca alloy region existed surrounding the RPs region and between the reacted β-TCP particles in the RPs area. EDS and AES confirmed the solid solution of Ca in the Mg matrix (i.e., Mg–Ca alloy region) in our previous study [24,34].

### 4.2. Degradation Behaviors of Sintered Mg and Mg/B-TCP Composites in a Physiological Saline Solution

The measured mass change consists of mass loss by degradation and mass gain by deposition of degradation products. Sintered Mg showed a preferential corrosion attack on the Mg particle boundaries, which led to cracks along them. Dezfuli et al. also reported that local corrosion occurred along particle boundaries for pure Mg produced by the sintering process [46]. The crack formation increased the area exposed to the solution and resulted in larger areas covered by flower-like porous degradation products. The results of XRD and EDS showed the degradation products were $Mg(OH)_2$. Consequently, the sintered Mg showed disintegration with a considerable mass increase after three days of immersion. The continuous mass increase indicated the porous degradation products did not prevent degradation effectively.

In contrast, Mg/β-TCP composites did not show such cracks as seen in sintered Mg because of the interfacial bonding enhanced by reactions during sintering. As our previous studies and other reports about Mg alloy/ceramic composites demonstrated [34,35,47], microgalvanic corrosion was observed for the pair of the Mg–Ca alloy region as an anode site and the neighboring non-alloyed Mg matrix as a cathode site. The anodic corrosion was observed as local corrosion including pitting on the Mg–Ca alloy region, adjacent to the RPs area. The rapid pH increment and its holding at around 10.5 was due to the corrosion on the anode sites and $Mg(OH)_2$ formation, respectively. Mg/20% β-TCP showed more rapid pH increment and rapid mass decrease than Mg/10% β-TCP, due to larger total areas of Mg–Ca alloy region as the anode sites and resultant more active anodic corrosion. Mg/10% β-TCP showed little mass change, possibly by balancing out mass gain due to deposition and mass loss by moderate microgalvanic corrosion. In addition, fewer pits on the Mg matrix of Mg/20% β-TCP than Mg/10% β-TCP were due to effective cathodic protection (Figure 4). Mg/β-TCP composites showed those microgalvanic behaviors in the revised simulated body fluids [34]. The stabilized mass change after three days for Mg/20% β-TCP was probably owing to the deposition of degradation layers such as $Mg(OH)_2$ on the anode sites, which suppressed the anodic corrosion. This is because the closely packed

plates of the degradation layers (Figure 7) potentially prevented direct exposure to the solution. In addition, the dense structure of the degradation products on the Mg/β-TCP composites improved corrosion resistance more considerably compared with the porous flower-like degradation products on the sintered Mg. Incorporation of Ca and P in the degradation layers may also promote corrosion resistance as Feng et al. demonstrated that the structure of degradation products of Mg alloy/calcium phosphate composites was denser and more protective than Mg alloy after immersion in physiological saline [14]. The different morphologies of degradation products may be the result of the different stages of degradation, as triangle-shaped degradation products of $Mg(OH)_2$ were seen in the very initial stage of degradation [48]. The region covered by triangle-shaped products was supposed to be the area showing metallic luster on the Mg matrix (Figure 4 (H1)), which demonstrates the effective cathodic protection functioning even after three days of immersion. The proposed degradation behaviors involving microgalvanic corrosion are unique to the Mg-based composites fabricated via reaction sintering [34,47]. Zeqin Cui et al. reported that nano-HA additives in Mg–Zn/HA composites dissolved and fell off, causing micropores during degradation [49]. We did not observe such disintegration due to the enhanced interfacial bonding by reaction sintering. Local corrosion may be mitigated by a homogeneous distribution of β-TCP in the fabrication steps, as Pushan Guo et al. proposed controlling ball milling conditions to distribute additives homogeneously to improve corrosion resistance of Mg/Nano-HA composites fabricated by ball milling and SPS [33].

The usage of physiological saline in our study for the simplified and accelerated immersion test showed a significant pH increase, which may show adverse effects on cells in vivo. However, a pH buffer system and calcium ions and phosphate ions, which promote the formation of a calcium phosphate layer in the body fluid, may suppress significant increases in pH in vivo. Indeed, our previous study using revised simulated body fluid showed a small increment of pH [34].

### 4.3. Deterioration of Strength Due to Degradation in a Physiological Saline Solution

The RPs in the Mg/β-TCP composites acted as reinforcement to enhance their strength [24]. Thus, for as-sintered samples, the Mg/β-TCP composites exhibited enhanced strength compared with sintered Mg, and the Mg/20% β-TCP, having a larger RPs area, showed more enhancement of strength than the Mg/10% β-TCP. The small stress increase from yield strength to maximum strength for Mg/20% β-TCP suggested that the strengthening mechanism was primarily controlled by composite strengthening rather than strain hardening. It should be noted that the enhanced interfacial bonding with the infiltrated Mg–Ca alloy regions in RPs area produced by unique microstructural evolution enabled even large agglomerations of RPs to contribute to reinforcement unlike other Mg matrix composites with agglomerations of additives [23,24,29]. Crack propagation through the RPs area demonstrates that the interfacial bonding was sufficiently strong to transfer the load from the Mg matrix to the RPs [50]. On the other hand, relatively weak interfacial bonding on the Mg particle boundaries for sintered Mg allowed crack propagation, which led to overall failure.

For as-immersed samples, sintered Mg further weakened the interfacial bonding along the particle boundaries by the preferable corrosion attack, which probably facilitated the crack initiation and propagation during the compression test.

In contrast, the Mg/10% β-TCP maintained their strength due to the improved corrosion resistance by the closely packed degradation products and absence of the "weak" spots where both crack propagation and preferential corrosion attack occur like the particle boundaries of the sintered Mg. Degradation after nine days of immersion in physiological saline, a more severe corrosion environment than the human body, resulted in a 15.8% reduction in yield strength to 133.1 MPa on average, which is comparable to human cortical bones [2]. A degradation test for longer periods in vivo or in a closer in vitro environ-

ment to the human body is required for further assessment as biodegradable load-bearing implants.

In the case of increasing the β-TCP fraction, thus increasing the RPs areas, although the initial strength was enhanced, the strength rapidly decreased by degradation as seen for Mg/20% β-TCP.

Less dense degradation products and larger RPs area on the Mg/20% β-TCP compared with the Mg/10% β-TCP potentially resulted in more severe microgalvanic corrosion adjacent to the RPs area. This can reduce the effective load transfer along the interface from the Mg matrix to the RPs area and then lower the contribution of composite strengthening, which is the main cause of strength enhancement. Therefore, a higher β-TCP fraction resulted in a more rapid decrease in both the maximum strength and yield strength. In addition, the hydration of MgO in the RPs and microgalvanic corrosion in the infiltrated Mg–Ca region in the RPs area may also weaken the strength of the RPs and the effect of the RPs as reinforcement to the composite strengthening. It may be interesting to control β-TCP fractions to mitigate the adverse effects of β-TCP on degradation while retaining positive effects such as strength enhancement.

Figure 11 shows comparisons of strength retention of Mg-based composites [38,39,51]. Strength retention was calculated by dividing the strength change by the initial ultimate compressive strength before degradation for ultimate compressive strength. The ultimate compressive strength before and after degradation is shown in Figure S4 in the Supplementary Materials. The loss of mechanical strength of Mg/10% β-TCP composites is comparable with other reported Mg-based composites. Interestingly, strength retention rate can be categorized into two groups by their rates: low strength retention group, including Mg/20% β-TCP and Mg-3 Zn/5 wt.%HA, and high strength retention group, including the others. It may be interesting to perform a systematic study to assess the mechanical integrity of Mg-based composites.

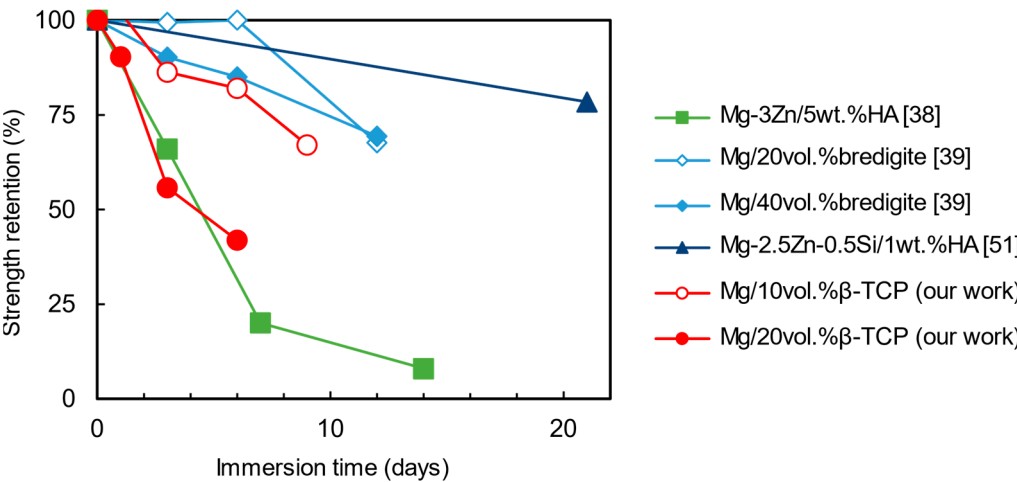

**Figure 11.** Strength retention of Mg-based composites before and after degradation.

## 5. Conclusions

Combination of immersion tests and post-mechanical tests for Mg/β-TCP composites having the evolved microstructure by advantageous reactions in the SPS with different β-TCP fractions (0, 10, 20 vol%) revealed effects of corrosion on the microstructure and strength retentions. The evolved microstructure underwent several counter-effects on mechanical integrity: (1) composite strengthening due to effective load transfer before corrosion, (2) elimination of "weak" spots which are subject to both mechanical crack propagation and preferential corrosion attack such as the boundaries of the sintered Mg particles, (3) rapid strength reduction due to severe microgalvanic corrosion and thus lowered effective load transfer, which was more prominent for a higher β-TCP fraction.

As a result, Mg/10% β-TCP demonstrated enhanced mechanical integrity: yield strength after nine days of immersion in physiological saline was compatible with the strength of human cortical bone. In vivo degradation tests for longer periods are still necessary to address clinical applications of Mg/β-TCP composites because bone-healing processes take 2–3 months and this work utilized simplified corrosion media. This study may facilitate the design of Mg/bioceramic composites fabricated via reaction sintering with the consideration of mechanical integrity.

**Supplementary Materials:** The following are available online at https://www.mdpi.com/2075-4701/11/2/227/s1. Figure S1: Optical microscopic image of Mg particles; Figure S2: Histogram of Mg particle size measured from optical microscope images; Figure S3: The boundary of sintered Mg particles obtained by Auger electron microscopy. (a) secondary electron image, (b) elemental mapping of Mg, and (c) elemental mapping of O; Figure S4: Ultimate compressive strength of Mg/bredigite, Mg-3Zn/5wt.%HA, Mg-2.5Zn-0.5Si/1wt.%HA, and Mg/β-TCP (our work); Table S1: Impurity of magnesium powder.

**Author Contributions:** Conceptualization, K.N. and E.K.; methodology, K.N.; investigation, K.N.; resources, E.K. and T.S.; writing—original draft preparation, K.N.; writing—review and editing, K.N., E.K., and S.H.; supervision, E.K. and T.S.; project administration, E.K.; funding acquisition, E.K. and T.S. All authors have read and agreed to the published version of the manuscript.

**Funding:** This work was financially supported by Light Metal Educational Foundation, Inc. The funding source does not involve study design.

**Institutional Review Board Statement:** Not applicable.

**Informed Consent Statement:** Not applicable.

**Data Availability Statement:** The data presented in this study are available on request from the corresponding author.

**Acknowledgments:** The authors would like to thank M. Ueda and K. Kawamura in Tokyo Institute of Technology for providing the SPS machine. They also acknowledge Y. Takayama in Tokyo Institute of Technology for helping fractographic observation by SEM. They also would like to thank V. Lambert for proofreading the manuscript.

**Conflicts of Interest:** The authors declare no conflict of interest.

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
