# Peer review of "Effects of Incorporating Β-Tricalcium Phosphate with Reaction Sintering into Mg-Based Composites on Degradation and Mechanical Integrity"

_metals, doi:10.3390/met11020227_

Round 1
Reviewer 1 Report
Authors deal with an interesting topic of biodegradable materials prepared by spark plasma sintering (SPS). The knowledge of reactions during spark plasma sintering is important in order to know the properties of the final product. This field is interesting as possible products might be metastable or just partly transformed due to the rapid sintering time compared to the conventional sintering [1-7]. The authors tried to use appropriate methods for characterisation of prepared materials. Nevertheless, in my opinion, the authors did a few errors in the design of experiments and the subsequent evaluation of data. Moreover, some results are not supported by appropriate measurements. A detailed description of potential errors and missing information is below. The beneficial part of the manuscript would be the measurement of the loss of mechanical properties with exposition time in a corrosion environment. However, the corrosion measurements could be elaborated better, and therefore, the measured values have almost no merit. Another benefit would be the description of the reaction between Mg and β-TCP which was already described better in previous manuscripts. The manuscript requires moderate revisions or the proper disproval of my concerns.
Line 110: What was the lower limit of particle size if known? Or the average size. What was the shape of the particles?
Line 112: Authors state that the purity of the powder was 99.5 %. Pure magnesium is very prone to impurities such as Fe, Cu, Ni, Co. Have the authors measured the amount of these impurities? If the amount of these elements is above a certain limit it has serious consequences on the corrosion resistance. Moreover, the powder was milled in a ball planetary mill – from what material were the balls and the container? The impurities usually increase after milling if milling materials are from steel [3]. Add the conditions of milling (RPM, time, ball/powder weight ratio). It may happen (in certain RPM, time, and ball ratio) that even after milling the reinforcement is homogeneously incorporated in the Mg matrix.
Line 132: Authors should consider that the mass change before and after immersion is affected by the difference between dissolved corrosion products and corrosion products precipitated on the surface. The true value of degradation rate should be calculated after removal of the corrosion products in the appropriate medium according to the ASTM-G31-72.
Line 156: The reaction product “RP” is not in my opinion necessarily entirely the product of the reaction between Mg and β-TCP. 1) MgO is normally on the surface of the powder even before sintering. 2) MgO can be created on the interface between individual Mg particles during sintering as well even under vacuum conditions due to the residual air between particles [7]. Authors should consider these two points as an additional source of MgO in the material.
Line 160: The EDS linescan of the interface is not enough evidence to claim that Ca diffused to the adjacent Mg matrix. The point analysis is probably affected by nearby β-TCP as P was detected there as well (The detection area of EDS can be about 2 µm). Moreover, claiming that C diffused from carbon die is bold. In which form would C be there? Dissolved/dispersed? Moreover, SPS is a very fast process while diffusion needs some time. It could not diffuse from the edge of the sample to its center. In my opinion, light elements such as C cannot be taken into account accurately unless standard materials are used for the calibration of EDS. I don’t know how accurate microscope authors were using, but the microscope at my disposal sometimes automatically detects C even in the casted Mg. Therefore, authors should consider their statements and edit them or provide more proof.
Fig.4: I recommend preparing a cross-section of the corroded products (See e.g. Fig. 5 in [8]). In order to do so, take the sample, mount it into epoxy resin, and grind it and polish it. Then authors should be able to see the progress of the corrosion front deep into the material. In pure Mg, there should be visible corrosion front attacking the particle interface [7]. In the case of composite materials, it could be interesting. It can be expected that the Mg-Mg boundary (MgO) will be attacked predominantly as well. Or it could spread in the Mg-β-TCP boundary as well?
Fig. 5: Authors should realise the effect of pH on the degradation rate of magnesium materials. Since the pH was about 10.5 after approximately 30 minutes of immersion. According to the Pourbaix diagram the Mg matrix is now in passivity [9]. Therefore after 30 minutes, the solution is no more aggressive (except Cl- ions). How do authors explain the rapid increase of pH in the case of composites while gradual in the case of pure Mg? The pH of the corrosion medium is a very important factor. Authors should either use a much larger volume of the medium or use buffers. Otherwise, important data can be obtained only before reaching passivity pH.
Line 240: Is that mole or weight ratio? And what should this ratio mean? Again, these values might be affected by the measured area beneath the corrosion products. Authors should realise that EDS measures also to a certain depth, and therefore, it could be affected by the β-TCP beneath the corrosion products.
Line 249: Why was pure Mg characterised with the remarkable mass increase while other samples remained constant or decreased? What does it mean?
Fig. 10: Again, I recommend cross-sections of the fracture in order to see if it goes along the particle boundary or transparticle etc…
Overall concerns, my opinions, and possible causes and solutions:
Please keep in mind I only see data in the manuscript and my claims below are just suggestions and they might be wrong due to the lack of data. Nevertheless, if my suggestions are wrong please discuss why.
The microstructure:
The description of the microstructure is short and a little confusing without the reading of the other two articles where the reaction between Mg and β-TCP is more visible. Nevertheless, in the discussion authors claim that MgO and Ca(PO3)2 are products of the reaction between Mg and β-TCP. Also, that Ca diffused from β-TCP into Mg without forming of Mg2Ca. Authors speak of Mg-Ca intermetallic compound / Mg-Ca molten region / Mg-Ca alloy and also the incorporation of Ca into MgO, so what is true? In which form is Ca? Authors used EDS linescan to show the diffusion of Ca – this is not the right method as mentioned above. Another proof is the XRD spectrum which is OK for MgO and Ca(PO3)2. The peak for β-TCP is so low because most of it was used for the reaction? Nevertheless, the form of Ca is still a question. In my opinion, Ca cannot be in the solid solution of Mg neither incorporated in MgO. According to thermodynamics, CaO is more stable than MgO [10], and therefore, the reaction MgO + Ca = CaO + Mg is more probable. Was CaO detected by XRD? The thermodynamic background of proposed reactions should be stated. And if it is not according to the thermodynamics then discuss why.
Corrosion:
The design of the corrosion experiment is wrong. The volume of 50 ml of medium is too low. According to the results, the pH will rise to 10.5 in about 30 minutes. Magnesium is passive in that high pH and only chloride ions disturb the passive layer of Mg(OH)2. Therefore, these results have no relevance and cannot say how this material would perform in the human environment at all. What in my opinion happened is: Sudden increase of pH in the case of composite materials is associated with rapid initial corrosion. The first places attacked were the boundaries between Mg particles as in the case of pure Mg and especially the boundaries between Mg and β-TCP which consist of MgO and probably CaO. And CaO is very reactive with water and forms Ca(OH)2 which rapidly increased pH and supported the passivation of the Mg matrix due to the pH of 10.5. The exposed surface area of the Mg matrix was also much lower compared to pure Mg. Moreover, this material's corrosion performance is very poor if it loses integrity after 9 days of exposition in a high pH solution. The biomaterials have to sustain their mechanical properties for at least 2-3 months. What would be the utilization of these materials in medicine?
Mechanical properties:
The authors measured compressive mechanical properties. Although the tensile properties would be interesting I understand that the measurement of tensile tests for such a small sample is problematic. A significant increase in compressive properties is associated with hard reinforcement. The most beneficial part of the paper would be the measurement of mechanical properties after exposition to the corrosion environment. Unfortunately, the loss of mechanical properties is high even for the best sample. Moreover, this data is associated with the corrosion measurements which were in my opinion poorly executed.
For the acceptance of the manuscript these points have to be discussed:
1 – The effect of impurities on the corrosion resistance.
2 – The effect of MgO on the corrosion resistance of magnesium materials
3 – In which form is Ca in the material.
4 – The thermodynamic background of the reaction.
5 – Passivation of Mg due to the high pH.
6 – Rapid increase of pH in the case of composites
7 – What utilization should these materials have in medicine?
8 – Comparison of mechanical and corrosion properties with other materials – better nor not
9 – Approve or disprove suggestions above
Optional points improving the quality of the manuscript but requiring re-measurements:
1 – Corrosion tests with at least 100 ml of the buffered medium on 1 cm2 of the sample in order to not reach passivity of Mg (pH = 10.5). Note that if using SBF – the threshold for pH is lower due to the possible precipitation of substances from the solution. And the measurement of the degradation rate after removal of the corrosion products in an appropriate medium.
2 – Cross sections of the sample after exposition in order to see where the corrosion front progress and how deep it is.
3 – Cross-sections or surface of the fractures – to see if it goes along particle boundaries etc…
Also, many style errors and typos were found:
Line 48: It seems there is extra space before „(130-180 MPa)“
Line 60: It seems there is extra space before „Among those…“
Line 63: I believe there is a typo „power metallurgy“ instead of powder metallurgy?
Line 70: It seems there is extra space before „In our previous study…“
Line 90: It seems there is extra space before „Jie Zhou’s group…“
Line 94: Typo “contrst” instead of contrast.
Line 125: It seems there is extra space before „Three specimens…“
Line 156: Why “(ref 20)” instead of [20]?
Line 157: It seems there is extra space before „…and (E)…“
Line: 186: missing dot at the end of the sentence.
Line 165: Typos: “Balancecd” Instead of balanced and “shwon” instead of shown extra space before “Mg was not…”.
Line 199: Typo “homogernously” instead of homogenously.
Line 252: It seems there is extra space before „The mass…“
Line 300: typo “aggolomorations“ instead of agglomerations
Line 301: typo: “ractions prouced” instead of reactions produced
Line 305: typo: “oxgen” and “lilkely” instead of oxygen and likely
Line 306: typo: “simulatnesouly” and “incorpolated” instead of simultaneously and incorporated
Line 307: typo: “intercoplation” instead of intertcalation
Lines 307 and 308: typos: “resulatnt expnasion of the interplaner distance were confimred" instead of resultant expansion of the interplanar distance were confirmed
Line 311: typo: “surronding” instead of surrounding
Through the manuscript there are sometimes missing spaces before references [X].
[1] Soderlind, J.; Cihova, M.; Schäublin, R.; Risbud, S.; Löffler, J. F., Towards refining microstructures of biodegradable magnesium alloy WE43 by spark plasma sintering. Acta Biomater. 2019, 98, 67-80.
[2] Cao, N. Q.; Narita, K.; Kobayashi, E.; Sato, T., Evolution of the microstructure and mechanical properties of Mg-matrix in situ composites during spark plasma sintering. Powder Metall. 2016, 59 (5), 302-307.
[3] Karasoglu, M.; Karaoglu, S.; Arslan, G., Mechanical properties of Mg-based materials fabricated by mechanical milling and spark plasma sintering. Proceedings of the Institution of Mechanical Engineers, Part L: Journal of Materials: Design and Applications 2019, 233 (10), 1972-1984.
[4] Minárik, P.; Zemková, M.; Lukáč, F.; Bohlen, J.; Knapek, M.; Král, R., Microstructure of the novel biomedical Mg–4Y–3Nd alloy prepared by spark plasma sintering. J. Alloys Compd. 2020, 819, 153008.
[5] Dvorsky, D.; Kubasek, J.; Vojtech, D., A new approach in the preparation of biodegradable Mg-MgF2 composites with tailored corrosion and mechanical properties by powder metallurgy. Mater. Lett. 2018, 227, 78-81.
[6] Viswanathan, R.; Rameshbabu, N.; Kennedy, S.; Sreekanth, D.; Venkateswarlu, K.; Rani, M. S.; Muthupandi, V., Plasma Electrolytic Oxidation and Characterization of Spark Plasma Sintered Magnesium/Hydroxyapatite Composites. Light Metals Technology 2013 2013, 765, 827-831.
[7] Dvorsky, D.; Kubasek, J.; Jablonska, E.; Kaufmanova, J.; Vojtech, D., Mechanical, corrosion and biological properties of advanced biodegradable Mg-MgF2 and WE43-MgF2 composite materials prepared by spark plasma sintering. J. Alloys Compd. 2020, 825, 154016.
[8] Ma, M.; Pokharel, D. B.; Dong, J.; Wu, L.; Zhao, R.; Zhu, Y.; Hou, J.; Xie, J.; Sui, S.; Wang, C.; Ke, W., In vivo corrosion behavior of pure magnesium in femur bone of rabbit. J. Alloys Compd. 2020, 848, 156506.
[9] Song, G. L.; Atrens, A., Corrosion Mechanisms of Magnesium Alloys. Adv. Eng. Mater. 1999, 1 (1), 11-33.
[10] Kim, Y. M.; Yim, C. D.; Kim, H. S.; You, B. S., Key factor influencing the ignition resistance of magnesium alloys at elevated temperatures. Scripta Mater. 2011, 65 (11), 958-961.
Reviewer 2 Report
Remarks in attached file.

Author Response
As reviewer 2's comments were directly annotated on the PDF of the manuscript, we addressed those points by revising our manuscript. The revised sentences are highlighted in yellow.
Reviewer 3 Report
-Why β-tricalcium phosphate? and just 10% and 20%? recommended 15%
-Need to compare with other related works fabricated by SPS in terms of mechanical and corrosion protection property
-How the SPS parameters being selected? small changes in SPS parameters like time and load and temp can vary the efficiency of the composite.
-Abst and Concl. need to be fully revised to provide comprehensive and precise info for readers.
- In Intro, some other works that being perofrmed in MG based SPS fabricated composites need to be reported :
- https://www.sciencedirect.com/science/article/pii/S1005030214001509
- https://www.sciencedirect.com/science/article/abs/pii/S0254058418302219
- https://www.sciencedirect.com/science/article/pii/S0921509320307401
- https://www.sciencedirect.com/science/article/pii/S0925838811005330
Round 2
Reviewer 1 Report
Authors significantly improved the quality of the manuscript. The detailed description of materials and methods was added. Although metastable and non-equilibrium phases are hard to describe, the proposed reaction products might be possible and are proved by relevant observations.
I still have reservations about the corrosion part. Authors oppose that the low volume of corrosion medium is similar to the body parts with a low amount of body fluid. However, the body environment is a dynamic and not stationary environment which allows ion and pH exchange. Otherwise, the sudden rise of pH around the implant would cause serious tissue damage. Therefore, it is more suitable to investigate corrosion processes in a buffered medium and with higher volume in order not to be saturated by Mg2+ ions. On the other hand, I understand that the authors wanted to avoid using an environment containing P and Ca in order to see the effect of TCP reinforcement. Authors are also right, that a high concentration of Cl- ions reveal the strengths and weaknesses of the material in a short time even in a high pH medium. Therefore, I agree that the proposed method is suitable for simplification and comparison of prepared materials among themselves, however, different behavior can be expected in human body simulated environment.
The part with mechanical properties was improved by the comparison with the literature (Fig. 11). This figure is very informative and useful, although different corrosion environments were used, which may distort the values.
I believe that the manuscript after submitted revisions and discussion is suitable for publication as it now contains a description of used methods, proper description of microstructure, comparable corrosion test, and measurement of mechanical properties after degradation with proper discussion and comparison with literature.